# Ingestion of *Helianthus tuberosus* at Breakfast Rather Than at Dinner is More Effective for Suppressing Glucose Levels and Improving the Intestinal Microbiota in Older Adults

**DOI:** 10.3390/nu12103035

**Published:** 2020-10-03

**Authors:** Hyeon-Ki Kim, Hanako Chijiki, Takuya Nanba, Mamiho Ozaki, Hiroyuki Sasaki, Masaki Takahashi, Shigenobu Shibata

**Affiliations:** 1Faculty of Science and Engineering, Waseda University, 2-2 Wakamatsu-cho, Shinjuku, Tokyo 162-8480, Japan; hk.kim@aoni.waseda.jp (H.-K.K.); hiroyuki-sasaki@aoni.waseda.jp (H.S.); 2Graduate School of Advanced Science and Engineering, Waseda University, 2-2 Wakamatsu-cho Shinjuku, Tokyo 1628480, Japan; hnk-1022@akane.waseda.jp (H.C.); n-x.t.x-n@asagi.waseda.jp (T.N.); mo_u2@ruri.waseda.jp (M.O.); 3Institute for Liberal Arts, Tokyo Institute of Technology, 2-12-1, Ookayama Meguro-ku, Tokyo 152-8550, Japan; takahashi.m.bp@m.titech.ac.jp

**Keywords:** *Helianthus tuberosus*, intestinal environment, postprandial glucose level, chrono-nutrition, circadian rhythm, glucose metabolism

## Abstract

To date, nutritional studies have focused on the total intake of dietary fiber rather than intake timing. In this study, we examined the effect of the timing of daily *Helianthus tuberosus* ingestion on postprandial and 24 h glucose levels, as well as on intestinal microbiota in older adults. In total, 37 healthy older adults (age = 74.9 ± 0.8 years) were recruited. The participants were randomly assigned to either a morning group (MG, *n* = 18) or an evening group (EG, *n* = 17). The MG and EG groups were instructed to take *Helianthus tuberosus* powder (5 g/day) just before breakfast or dinner, respectively, for 1 week after the 1-week control period. The glucose levels of all participants were monitored using a continuous glucose monitoring system throughout the 2 weeks. The intestinal microbiota was analyzed by sequencing 16S rRNA genes from feces before and after the intervention. There were no significant differences in the physical characteristics or energy intake between groups. *Helianthus tuberosus* intake led to decreases in tissue glucose levels throughout the day in both groups (*p* < 0.01, respectively). As a result of examining the fluctuations in tissue glucose levels up to 4 hours after each meal, significant decreases in the areas under the curves (AUCs) were observed for all three meals after intervention, but only in the MG (breakfast: *p* = 0.012, lunch: *p* = 0.002, dinner: *p* = 0.005). On the other hand, in the EG, there was a strong decrease in the AUC after dinner, but only slight decreases after breakfast and lunch (breakfast: *p* = 0.017, lunch: *p* = 0.427, dinner: *p* = 0.002). Moreover, the rate of change in the peak tissue glucose level at breakfast was significantly decreased in the MG compared to the EG (*p* = 0.027). A greater decrease was observed in the change in the blood glucose level after the ingestion of *Helianthus tuberosus* in the MG than in the EG. Furthermore, the relative abundance of *Ruminococcus* in the MG at the genus level was significantly higher at baseline than in the EG (*p* = 0.016) and it was also significantly lower after the intervention (*p* = 0.013). Our findings indicate that *Helianthus tuberosus* intake in the morning might have relatively stronger effects on the intestinal microbiota and suppress postprandial glucose levels to a greater extent than when taken in the evening.

## 1. Introduction

The increasing prevalence of diabetes and its impact on morbidity and mortality have become an important public health problem. In 2016, the World Health Organization (WHO) estimated that almost 422 million adults live with diabetes; this is an increase in global prevalence from 4.7% in 1980 to 8.5% in 2014 [1]. Similarly, the number of diabetes patients in Japan is increasing rapidly. Aging is also associated with an elevated risk of diabetes, which is likely related to a sedentary lifestyle and age-related decline in glucose tolerance and insulin function [2,3,4].

The association between postprandial hyperglycemia and the risk of type 2 diabetes and cardiovascular disease (CVD) is well known in humans [5,6,7]. Thus, it is important to suppress postprandial hyperglycemia from the viewpoint of prevention of lifestyle diseases. Glucose tolerance has been reported to be regulated by a circadian system, including time of day variations in digestion, absorption, and metabolism in the stomach and intestines [8,9]. Glucose tolerance, including insulin function, is highest in the morning and is lowest in the evening and night [10]. Postprandial glucose levels have also been suggested to be higher in the evening than in the morning [11,12,13]. In other words, there is a possibility that glucose metabolism function decreases at dinner time. Therefore, it has been indicated that meal timing is an important factor that regulates glucose tolerance, including insulin sensitivity.

In addition, dietary fiber is an indigestible food ingredient that exerts a beneficial effect on the host by suppressing the growth of useful bacteria or harmful bacteria that reside in the large intestine [14]. It has been shown to improve the blood glucose level and reduce the risk of lifestyle-related diseases such as diabetes and obesity [14]. In particular, water-soluble dietary fiber has been found to suppress the increase in blood glucose levels by slowing the absorption of glucose [15]. Dietary fiber also improves intestinal environment. Short-chain fatty acids (SCFAs) are the end of products of anaerobic bacterial fermentation and they lower intestinal pH, suppress the growth of pathogenic bacteria in the gut, and regulate immune and metabolic functions [16,17]. Furthermore, glucose metabolism has been shown to be associated with the intestinal environment, and it has been reported that the intestinal microbiota is involved in major metabolic functions such as glucose regulation and insulin resistance [18,19,20].

Inulin is one of the polysaccharides produced from many plants and is abundantly contained in *Helianthus tuberosus* (i.e. Jerusalem artichoke), burdock, and chicory [21]. Previous studies have shown that inulin intake alters the intestinal microbial composition and increases the number of beneficial bacteria, such as *bifidobacteria*, in feces [22,23]. Until now, nutritional studies have focused on total dietary fiber intake rather than intake timing. Circadian variation in the intestinal microbiota has been shown to be controlled by dietary composition [24,25]. Thus, some food ingredients are more effective depending on when they are consumed. In fact, some previous studies have reported that acute and consecutive intake of catechin-rich green tea in the morning or evening has different glucose-lowering effects in humans [26,27]. Moreover, in mice, inulin ingestion in the morning has a greater effect on microbiota diversity than when it is ingested in the evening [28]. In addition, it has been reported that *Helianthus tuberosus* intake has an effect on the intestinal microbiota in humans [29].

However, the effect of *Helianthus tuberosus* intake in the morning or evening on blood glucose levels and intestinal microbiota in humans remains unclear. Therefore, the present study aimed to investigate the effect of daily ingestion of powdered *Helianthus tuberosus* in the morning or evening on glucose levels and intestinal microbiota.

## 2. Materials and Methods

### 2.1. Study Participants

This study included healthy older adults (*n* = 37; 18 men and 19 women) who were ≥65 years old in Tokyo (Japan). The study was conducted between July 2018 and September 2018, and the inclusion criteria were as follows: (1) not taking any antioxidant, anti-obesity, or anti-diabetes supplements; (2) no diagnosis of diabetes, dyslipidemia, or sleep apnea syndrome by a doctor; (3) hypertension (systolic blood pressure: >140 mmHg, diastolic blood pressure: >90 mmHg); and (4) absence of the use of glucose/insulin-lowering or related medications. The study protocol conforms to the Helsinki Declaration and was approved by the Ethics Committee for Humans at Waseda University (approval number: 2018-031). The human trial was registered at UMIN (www.umin.ac.jp/ctr/ number: UMIN000032858). Informed consent was obtained from all participants after explaining the experiment details to them. 

All participants answered a questionnaire on dietary habits, lifestyle habits, and health and medication status prior to the study. Participants were excluded from the study for the following reasons: did not meet the inclusion criteria (*n* = 2), had glucose monitoring sensor removed (*n* = 2) or failure (*n* = 1), and submersion of feces (*n* = 8). Consequently, thirty participants were included in the analysis (Figure 1).

### 2.2. Study Design

A single-blind, parallel design was used. The subjects were randomly assigned to the following two groups: the morning intake group (MG, *n* = 18; men, *n* = 9; women, *n* = 9) and the evening intake group (EG, *n* = 17; men, *n* = 8; women, *n* = 9). The experiment was conducted for 2 weeks. We defined the first week (Days 1 to 7) as the “Baseline” period and asked the participants to maintain their normal life without changing lifestyle habits such as diet and exercise. In the second week (Days 8 to 14), described as the “Intervention period”, subjects were asked to maintain their same daily life with dietary contents that were as similar to the first week as possible. Thus, the effects of probiotics from meals and antibiotics may be minimized in the current experiment. It has previously been shown that inulin intake for 7 days changes the intestinal environment in humans [30]. Therefore, it is considered that the effect of intake timing on the intestinal environment can be sufficiently examined in a week. Additionally, 5 g of *Helianthus tuberosus* powder (Kumamoto Prefecture, Hinata’s *Helianthus tuberosus*) in normal/cold water or hot water was taken just before each breakfast or dinner. In previous studies, an intake of dietary fiber of approximately 5–20 g/day was used [31]. Therefore, in this study, the powder intake was set to 5 g in consideration of reducing the burden on the elderly participants.

During the baseline period, the participant’s physical characteristics were measured, and participants filled out a lifestyle survey (e.g., mealtime, chronotype) and a food frequency survey. During the two-week experiment period, all subjects were asked to evaluate their physical activity level with a triaxial accelerometer, and to wear a continuous glucose monitoring system. They were also asked to collect their feces in a tube with phosphate-buffered saline containing 20% glycerol, which had been distributed in advance, before and after the intervention period for intestinal microbiota evaluation (mornings of Day 8 and 15). Collected fecal samples were transported to the laboratory at 4 °C and were then immediately frozen in liquid nitrogen and stored at −80 °C until analysis.

### 2.3. Helianthus tuberosus Contents

The *Helianthus tuberosus* powder used in this study was produced at Aso Shizen no Megumi Souhonpo Co., Ltd, Kumamoto, Japan. On a dry weight basis, *Helianthus tuberosus* contains approximately 60% water-soluble dietary fiber. All MG and EG participants were issued a package containing a 1-week supply of *Helianthus tuberosus* powder (5 g) and they were told to take it just before breakfast or dinner during the intervention period of 1 week.

### 2.4. Measurements

#### 2.4.1. Anthropometry

Body mass was measured to the nearest 0.1 kg using a digital balance (Inbody 230, Inbody Inc., Tokyo, Japan) and height was measured to the nearest 0.1 cm using a wall-mounted stadiometer (YS-OA, As One Corp., Japan). Body mass index (BMI) was calculated as weight in kilograms divided by the square of height in meters, while muscle mass was measured by direct segmental multifrequency (20 kHz to 100 kHz) bioelectrical impedance (Inbody 270, Inbody Inc., Tokyo, Japan).

#### 2.4.2. Tissue Glucose Level and Analysis

All subjects were required to wear a continuous glucose monitoring system (Continuous Glucose Monitoring System, FreeStyle Libre Pro, Abbott Laboratories, Chicago, IL, US) for the continuous measurement of tissue glucose levels during the intervention. Once applied, the system can continuously measure tissue glucose levels in 15 min intervals for 14 days and it is thought to be less burdensome, even for elderly people. The sensor was applied to the back of the upper arm. The sensor continually stores data on tissue glucose levels. In both groups, the tissue glucose level and area under the curve (AUC) for 24 h after the *Helianthus tuberosus* intake and 4 h after the intake of breakfast, lunch, and dinner were calculated. In addition, the parameters used for the evaluation of glycemic variability were the standard deviation (SD) of glucose; coefficient of variation (CV); maximum glucose level (MAX); minimum glucose level (MIN); and mean amplitude of glycemic excursion (MAGE). The SD and MAGE values were calculated from 2400 h to 2400 h the next day while MAX and MIN were taken as the highest and lowest glucose values, respectively.

#### 2.4.3. Physical Activity Assessment

All participants were asked to wear a triaxial accelerometer (Active style Pro HJA-750C, Omron Corp., Kyoto, Japan) for a week. They wore the accelerometer each day at all times from morning until evening except during shower times. The device determined the level of intensity (METs) generated by activity every 10 seconds from 0–8 METs (where 0 was the lowest and 8 was the highest). Data from the participants, who wore the accelerometer for at least 10 h (600 min) daily for at least four weekdays and a day on the weekend, were included. Apart from that, the duration of daily moderate to vigorous physical activity (MVPA) was calculated and used to estimate weekly activity by calculating a weighted average of daily weekday and weekend activities (i.e. weekly MVPA = [average daily weekday MVPA × 5] + [average daily weekend MVPA × 2]). All minute recordings ≥3 METs were classified as MVPA.

#### 2.4.4. Chronotype Assessment

Chronotype was evaluated using the morningness-eveningness questionnaire (MEQ) [32]. The MEQ is composed of 19 items related to sleep habits, sleepiness, and the preferred time for daily performance. The sum gives a score ranging from 16 to 86. Based on their scores, the participants were divided into the following three-category classification: morningness (score 59–86), intermediate (score 42–58), or eveningness (score 16–41).

#### 2.4.5. Dietary Assessment

The food frequency questionnaire (FFQ) is one of the most commonly used evaluation methods for meals. Most FFQs for Japanese people are highly effective for estimating nutrients [33]. The questionnaire examines the content of meals using simple questions involving 29 food groups and 10 types of cooking methods, divided according to the food group. Average daily energy intake was depicted as kilocalories per day (kcal/day). Total dietary fiber was described as grams per day (g/day).

#### 2.4.6. Fecal pH Measurement

Fecal pH was evaluated using a pH meter (pH Spear; Eutech Instruments, Vernon Hills, IL, USA). First, the frozen feces were crushed and a 20 mg sample was collected. Then, 1 mL of PBS was added and homogenized. The mixture was then centrifuged at 12,000 rpm for 15 min at 4 °C and the supernatant was measured with a pH meter.

#### 2.4.7. Short-Chain Fatty Acid (SCFA) Measurement

SCFA in fecal contents was evaluated through gas chromatography (Shimadzu Co., Kyoto, Japan) as described previously [34]. We collected 1 g of feces, then added 8 mL of ether (FUJIFILM Wako Pure Chemical Co., Osaka, Japan) and 4 mL of ethanol (FUJIFILM Wako Pure Chemical Co., Osaka, Japan). After, we added 1 mL of sulfuric acid to extract SCFAs by stirring. Next, we centrifuged the samples at 9000 rpm for 1 min at room temperature. Then, we picked up the supernatant and injected it into a capillary column (InertCap Pure-WAX (30 m × 0.25 mm, df = 0.5 µm); GL Sciences, Tokyo, Japan). A standard curve was plotted to quantify the amount of each acid in the samples. 

#### 2.4.8. Fecal DNA Extraction and 16S rRNA Gene Sequencing

16S rRNA gene sequencing was performed according to the Illumina instructions. V3-V4 variable regions of the 16S rRNA gene were amplified by PCR using the following primers: 

F-5′-TCGTCGGCAGCGTCAGATGTGTATAAGAGACAGCCTACGGGNGGCWGCAG-3′

R-5′-GTCTCGTGGGCTCGGAGATGTGTATAAGAGACAGGACTACHVGGGTATCTAATCC-3′ 

Amplicon PCR was performed with 2.5 µL of microbial DNA (5 ng/µL), 5 µL of each primer (1 µM), and 12.5 µL of 2x KAPA HiFi HotStart Ready Mix (Kapa Biosystems, Wilmington, MA, USA). The cycling parameters were as follows: one denaturation cycle of 3 min at 95 °C, 25 cycles of 95 °C for 30 s, 55 °C for 30 s, and 72 °C for 30 s, as well as a final extension of 72 °C for 5 min. The PCR amplicons were purified using AMPure XP beads (Beckman Coulter, Inc., Brea, CA, USA).

To perform multiplex sequencing, adapters and barcodes were attached to amplicons using the Nextera XT Index Kit v2 (Illumina Inc., San Diego, CA, USA). Index PCR was performed with 5 µL PCR production, 5 µL of each Nextera XT Index primer, 25 µL of 2 × KAPA HiFi HotStart Ready Mix, and 10 µL of PCR Grade water under conditions of 3 min at 95 °C, eight cycles consisting of denaturation at 95 °C for 30 s, annealing at 55 °C for 30 s, and extension at 72 °C for 30 s, followed by a final extension at 72 °C for 5 min. The quality of the purifications was evaluated using an Agilent 2100 Bioanalyzer with a DNA 1000 kit (Agilent Technologies, Santa Clara, CA, USA). Finally, the DNA library was diluted to a concentration of 4 nM. The DNA library was sequenced using the Miseq Reagent Kit v3 (Illumina Inc.) in the Illumina Miseq 2 × 300 bp platform, according to the manufacturer’s instructions.

#### 2.4.9. Analysis of 16S rRNA Gene Sequences

The 16S rRNA sequence reads were processed by the quantitative insights into microbial ecology (QIIME) pipeline version 1.9.1 [35]. The quality-filtered sequence reads were assigned to operational taxonomic units (OTUs) using closed-reference OTU picking at 97% identity with the UCLUST algorithm [36]. These reads were then compared with the reference sequence collections in the Greengenes database (August 2013 version). In total, 1,296,946 reads were obtained from 44 samples. On average, 22,361 ± 2721 reads were obtained per sample. A taxonomy summary at the phylum level, beta-diversity (between-sample dissimilarity), and principal coordinate analysis (PCoA) were calculated by the QIIME software. PCoA analysis was also calculated using weighted UniFrac distances.

#### 2.4.10. Constipation Assessment scale (CAS) Assessment

The CAS was developed to enable the prediction of the risk of developing constipation. It contains 8 items and participants can achieve a maximum of 16 points. A score of 5 or more was considered to indicate constipation [37].

### 2.5. Statistical Analysis

Data were analyzed using Predictive Analytics Software for Windows (SPSS Japan Inc. Tokyo, Japan). Based on the distribution of postprandial glucose values in our previous study [38], the total sample size was calculated to be able to detect a medium effect (Cohen’s d (effect size) = 0.81). A total sample size of 27 was required to have approximately 80% power to detect large effects at a significance level of 0.05 (G*Power, version 3.1.9.2, Universitat Kiel, Germany). All parameters were tested for normal or non-normal distributions using the Kolmogorov–Smirnov test. To compare changes in the diurnal glucose level and postprandial glucose level between trials in both groups, a two-way repeated measures analysis of variance was used (effects of the trial and time were used as factors). When significant interaction effects were detected, we used the Bonferroni method for post-hoc comparisons. Pearson’s product-moment correlation coefficient was calculated to determine the relationship between changes in the blood glucose level and changes in the intestinal microbiota in both groups. To investigate changes in the blood glucose levels from baseline to the post-intervention period, we used an unpaired student’s *t*-test. If the data showed a non-normal distribution, statistical significance was determined using Wilcoxon’s test. *p*-values <0.05 were considered to indicate statistical significance. A permutational multivariate analysis of variance (PERMANOVA) was used to examine the changes in the microbiota composition and was performed using QIIME.

## 3. Results

### 3.1. Baseline Characteristics of Study Participants

There were no significant differences in the physical characteristics or energy intake between the groups (Table 1). In addition, FFQ data revealed that there were no significant differences in the intake of yogurt, milk, or cheese products at baseline between the two groups. Also, the intake rate of Helianthus tuberosus powder in this study was 100% in participants.

### 3.2. Physical Activity Level and Mealtime

There were no significant differences in physical activity levels between baseline and intervention in either group. From this, the effect of physical activity on tissue glucose levels was equivalent. In addition, there were no significant differences in mealtime between baseline and intervention. 

### 3.3. Comparison of the 24 h Tissue Glucose Level

In both groups, the consumption of Helianthus tuberosus powder decreased tissue glucose levels for 24 hours (Figure 2A,C). On the other hand, significant trial × time interactions were found for changes in the blood glucose levels after the intake of Helianthus tuberosus, but only in the MG (*p* = 0.047) (Figure 2A). Moreover, the AUC showed a significant decrease after intervention as compared with baseline (respectively: *p* < 0.01) in both groups (Figure 2B,D). In addition, we compared the 24 h tissue glucose level in men and women. The AUC of the 24 h tissue glucose level in men was significantly decreased after the intervention as compared with baseline, and AUC of the 24 h tissue glucose level in women tended to be insignificantly decreased after the intervention as compared with baseline. Conversely, the glucose parameters were not significantly different between the two groups (Table 2).

### 3.4. Comparison of the Tissue Glucose Level 4 h Postprandial

To evaluate fluctuations in more detail, we confirmed tissue glucose fluctuation up to 4 hours after each meal. Significant trial × time interactions were found for changes in the tissue glucose levels after the breakfast and lunch, but only in the MG (*p* = 0.019, *p* = 0.023) (Figure 3A,B). Furthermore, according to the AUC, in the MG, the tissue glucose level decreases after intervention for all three meals (breakfast: *p* = 0.012, lunch: *p* = 0.002, dinner: *p* = 0.005) (Figure 3A–C). On the other hand, in the EG, the decreases in tissue glucose levels after breakfast and lunch were smaller than that for dinner, though a decline was seen for all three meals (breakfast: *p* = 0.017, lunch: *p* = 0.427, dinner: *p* = 0.002) (Figure 3D–F). When comparing men and women, we found that in men there were significant trial × time interactions with tissue glucose fluctuation up to 4 hours after breakfast in the MG (*p* = 0.029), and the tissue glucose level after all three meals decreased significantly or tended to decrease after intervention as compared with baseline (breakfast: *p* = 0.060, lunch: *p* = 0.016, dinner: *p* = 0.007). On the other hand, in the EG, the tissue glucose level decreased after intervention, but only for breakfast, as compared with baseline (*p* = 0.024). Furthermore, in women, the tissue glucose level after lunch tended to decrease after the intervention in the MG (*p* = 0.059), and the tissue glucose level after dinner was significantly decreased in the EG (*p* = 0.002). We also compared the rate of change in the peak tissue glucose level after each meal in both groups. The rate of change in peak tissue glucose level at breakfast was significantly decreased in the MG compared to the EG (*p* = 0.027) (Figure 4A).

### 3.5. Comparison of SCFA and pH

There were no significant changes observed in total SCFA, acetic acid, propionic acid, or butyric acid in either group (Figure 5A,B,C,E). However, there was a significant increase in lactic acid after the intervention compared to baseline, but only in the MG (*p* = 0.013) (Figure 5D). There was no significant difference in pH between the morning and EGs (Figure 5F). Moreover, we calculated SCFAs and pH separately for men and women, but there were no significant changes between men and women.

### 3.6. Relationship between Changes in Tissue Glucose Levels and Changes in Intestinal Microbiota

In order to evaluate how the intestinal microbiota changed in people whose glucose level changed, we calculated the correlation between the change rate of the AUC of the daily glucose level and the amount of change in the intestinal microbiota relative abundance. Analysis of intestinal microbiota was carried out for 22 participants for whom we were able to collect both glucose level and fecal data. In this study, we focused on the Bacteroidetes and Firmicutes phyla, which are beneficial bacteria that are predominately found in the human gut. For the 22 participants, we found that the change rates of the AUC of the daily glucose level were negatively correlated with the relative abundance of phylum Bacteroidetes (*r* = −0.525, *p* = 0.012), and positively correlated with that of phylum Firmicutes (*r* = 0.539, *p* = 0.010) (Figure 6A,C). Comparing each group, the change rate of the AUC of the daily glucose level tended to be negatively correlated with the relative abundance of phylum Bacteroidetes (morning: *r* = −0.583, *p* = 0.077, evening: *r* = −0.530, *p* = 0.076) and positively correlated with that of phylum Firmicutes (morning: *r* = 0.603, *p* = 0.065, evening: *r* = 0.519, *p* = 0.084) (Figure 6B,D).

### 3.7. Comparison of the Relative Abundance of some Bacteria 

The relative abundance of *Ruminococcus* in the MG at the genus level was significantly higher at baseline than in the EG (*p* = 0.016) and significantly lower after the intervention (*p* = 0.013). However, no significant changes were observed in other bacteria (Table 3). Furthermore, we examined some bacteria separately for men and women, but there were no significant changes as a result of Helianthus tuberosus powder intake.

We also analyzed the PCoA of the weighted UniFrac distances and assessed the beta-diversity of the microbiota composition (Figure 7). However, there were no significant changes in the beta-diversity of the microbiota before and after the intervention in either group.

### 3.8. Comparison of the CAS

All participants defecated every morning. We compared the CAS before and after Helianthus tuberosus powder intake in the morning and evening. There were no significant changes in CAS in either group (Figure 8A). On the other hand, as a result of examining only the participants diagnosed with constipation (*n* = 7), there was a tendency of improvement in CAS after the intervention compared with the baseline (*p* = 0.088) (Figure 8B). Additionally, only the MG showed a trend towards improved CAS after the intervention (*n* = 4, *p* = 0.066) (Figure 8B).

## 4. Discussion

This study is one of the first to investigate the effects of timing of daily *Helianthus tuberosus* intake on postprandial glucose levels and intestinal microbiota in older adults. The main findings of our study are as follows. First, after a week of consecutive *Helianthus tuberosus* ingestion in the morning and evening, the changes in the tissue glucose level after each meal were greater and decreased in the morning than in the evening. Second, delta change of the AUC of the daily glucose level before/after intervention was negatively correlated with a relative abundance of phylum *Bacteroidetes* and positively correlated with that of phylum *Firmicutes*. Moreover, only the MG had increased lactic acid and a decreased relative abundance of *Ruminococcus* at the genus level. Third, it was shown that the ingestion of *Helianthus tuberosus* in the morning might be more preferable for improving constipation. Therefore, the present study suggested that *Helianthus tuberosus* ingestion in the morning has favorable effect on the postprandial glucose level and the intestinal microbiota as compared with the *Helianthus tuberosus* ingestion in the evening.

The effect of *Helianthus tuberosus* powder on glycemic control differs according to the timing of intake. In this study, morning intake was shown to be more effective than evening intake for controlling postprandial and daily glucose levels. Both morning and evening intake showed significant decreases in the 24 h blood glucose level. Conversely, the postprandial blood glucose level decreased significantly in the tissue of the morning intake group for all three meals (breakfast, lunch, and dinner), whereas in the evening intake group, a significant decrease in the blood glucose level was observed for two meals only (dinner and breakfast). This is likely due to the difference in the fasting time until the next meal. If the *Helianthus tuberosus* powder was taken at breakfast, the next meal would be lunch and then dinner on the same day, and the three meals would be consumed within approximately half a day. In addition to fasting time, a circadian variation of metabolic response may be involved in breakfast and dinner. Glucose tolerance has been reported to be regulated by a circadian system, including time of day variations in digestion, absorption, and metabolism in the stomach and intestines [8,9]. Furthermore, a previous study has shown that circadian variations are also present in the intestinal microbiota composition [24,25]. Therefore, the fasting period and circadian rhythm cooperatively caused the timing effect such as breakfast and dinner.

However, if the *Helianthus tuberosus* powder was taken for dinner, the next meal would be breakfast and then lunch on the following day. As such, the second meal would be taken approximately half a day after the *Helianthus tuberosus* powder was taken. Therefore, it is likely that when taken in the morning, the fasting time between the next meal and the meal afterwards would be shortened. Therefore, the second meal effect was observed, and the tissue glucose level was found to be decreased even after meals (lunch and dinner) without the addition of the *Helianthus tuberosus* powder. The second meal effect is when taking a meal that moderates blood sugar level increase for the first meal suppresses the blood sugar level increase at the second meal [39,40]. We recently published the second meal effect by dietary fiber intake with snacks [38]. Thus, dietary fiber rich food provides the second meal effect. However, the mechanism of the second meal effect has not yet been fully clarified. One mechanism is considered to involve free fatty acids, which are released on prolonged fasting, thereby causing insulin resistance. With consumption of a meal that moderates the increase in blood glucose, glucose is absorbed slowly, thus reducing the release of free fatty acids. Then, the next meal can be taken with good insulin sensitivity [41,42]. Therefore, an increase in the blood glucose level at the next meal can be suppressed. In addition, we also compared men and women regarding the effect of different intake times of *Helianthus tuberosus* powder on postprandial blood glucose fluctuations. The results for men were similar to the overall results, but no clear differences were seen for women. These differences between men and women are thought to be due to the different glycemic responses to diet [43]. In fact, the increases in tissue glucose levels after each meal in this study were greater in men than in women. Further, women tended to consume more dietary fiber per day than men (men: 16.8 g ± 2.5, women: 19.1 g ± 1.2, *P* = 0.061). Therefore, the contribution of *Helianthus tuberosus* powder intake on postprandial blood glucose levels in women may have been small. However, further studies will be needed to clarify the differences between men and women in more detail.

Another possible mechanism is the promotion of glucagon-like peptide-1 (GLP-1) secretion in the intestinal tract by increasing SCFAs. SCFAs are important for health because they improve the immune function of the large intestine as well as the energy metabolism in the liver and muscles [17]. SCFAs are also one of the most characteristic triggers for GLP-1 secretion [44]. GLP-1 is secreted by intestinal L cells and its enhanced secretion has been shown to be beneficial in diabetes patients [44]. Previous studies have reported that infusions of GLP-1 mimetics are associated with improved glycemic control and reduced incidence of hypoglycemia [45]. However, the total SCFA concentration in this study did not significantly change from before to after the intervention in either group. Therefore, it may be unlikely that the increase in SCFA concentration affects glycemic control. However, lactic acid was significantly increased, but only in the morning intake group. Lactic acid, an organic acid, is a known precursor of both acetic acid and butyric acid [46,47]. Considering the ratio of acetic acid and butyric acid to SCFAs, an increase in the concentration of lactic acid may lead to increased SCFAs in the long run. However, the fecal lactic acid concentration in this study was quite low, and a longer-term study is needed to determine whether this change contributes to the increase in SCFAs. In addition, because incretin (GLP-1 and glucose-dependent insulinotropic polypeptide, GIP) and insulin sensitivity have not been studied, the mechanism of action cannot be examined. Therefore, a more detailed study is needed in the future.

Morning intake of *Helianthus tuberosus* powder may be more effective in improving the intestinal environment. A meta-genomic analysis of healthy subjects revealed that almost all bacteria in the large intestine and the rectum belong to two phyla: *Bacteroidetes* and *Firmicutes* [48]. In this study, we investigated the relationship between changes in blood glucose levels and gut microbiota in the elderly who took *Helianthus tuberosus* powder. In all subjects, a negative correlation was seen between blood glucose changes and the phylum *Bacteroidetes*, and a positive correlation was found with the phylum *Firmicutes*. In addition, comparison between groups showed negative and positive correlation tendencies with the *Bacteroidetes* and *Firmicutes* phyla, respectively. The bacteria of the phylum *Bacteroidetes* were reported to be decreased and those of the phylum *Firmicutes* were increased in patients with type 2 diabetes and obesity [49,50]. Therefore, it is generally considered that increases in the phylum *Bacteroidetes* and decreases in the phylum *Firmicutes* are good changes for intestinal flora. This is because an increase in bacteria in the *Firmicutes* phylum contributes to the development of obesity by increasing energy absorption efficiency and is involved in increasing insulin resistance [51]. The results suggest that the phylum *Bacteroidetes* increased and the phylum *Firmicutes* decreased in subjects whose blood glucose level decreased due to the intake of the *Helianthus tuberosus* powder, regardless of the timing of intake. Furthermore, it was confirmed that more favorable changes occurred in the morning intake group than in the evening intake group. Compared to the evening intake group, the morning intake group had a greater number of plots with increased *Bacteroidetes* and decreased *Firmicutes* in response to a decreased blood glucose level. Therefore, morning intake of the *Helianthus tuberosus* powder may be more effective due to changes in the constituent bacteria of the intestinal flora. In our previous study with mice, a consistent finding has been obtained on the intake of inulin that is abundant in *Helianthus tuberosus*. However, *Bacteroidetes* and *Firmicutes* are inhabited by microbes with various functions. As such, this change cannot be simply asserted as a good change. Therefore, we investigated changes at the genus level depending on the timing of intake of the *Helianthus tuberosus* powder. However, the results showed that only the relative amount of *Ruminococcus* was higher before the intervention than in the evening intake group, but this amount was significantly lower after the intervention. Thus, the change of phylum levels of Bacteroidetes and Firmicutes could not be explained by genus levels of the microbiota. Because intestinal microbiota may fluctuate due to the intake timing of Helianthus tuberosus, a more detailed examination such as feces collection in the evening is required.

Morning intake of the *Helianthus tuberosus* powder improves constipation symptoms. Previous studies have shown that water-soluble dietary fiber has the effect of suppressing the increase in the blood glucose level and promoting the growth of beneficial bacteria by serving as a nutrient source [52]. As such, it has been shown to contribute to the maintenance of a good intestinal environment. Conversely, insoluble dietary fiber has been reported to contribute to the improvement of bowel movements by increasing the amount of feces and enhancing intestinal peristalsis [53]. Among those with constipation symptoms in this study, the improvement tendency was observed only with morning intake of *Helianthus tuberosus*. It was likely influenced by the fact that both soluble and insoluble dietary fibers present in *Helianthus tuberosus* promoted intestinal peristalsis because soluble fibers help to increase bacteria growth and insoluble fibers help in bowel movement. The gastrocolic reflex, where the intestines move after a meal, is most often triggered when food enters a fasting stomach and is therefore more likely to occur at breakfast [54,55]. All subjects in this study consumed breakfast, and it is likely that the conditions were similar between the two groups. Therefore, the intake of the *Helianthus tuberosus* powder in the morning likely promoted the peristalsis of the intestine due to the gastrocolic reflex.

The present study has some limitations. First, the participants in our study were healthy older adults. Our findings do not apply to other individuals such as healthy young men and women, or populations with diabetes. However, few healthy aging studies have been done on healthy older adults. Thus, the results of this study are considered to be important in that they may lead to future studies. Second, in this study, the test meals such as breakfast, lunch, and dinner were not controlled during the study period. Therefore, the effect of differences in meal intake on postprandial blood glucose levels and intestinal microbiota changes cannot be denied. However, participants were instructed to maintain their normal lifestyle during the study. In fact, there was no significant change in the total energy intake at baseline and after the intervention. Thus, the effect of differential meal intake on our results is likely to be weak. In future, it will be necessary to conduct studies that provide the same meal to all participants. Finally, the sample size in this study was small, thus making it difficult to draw conclusions about the effects of Helianthus tuberosus on postprandial glucose levels and intestinal microbiota. Particular attention should be paid to the interpretation of the effect of *Helianthus tuberosus* on the CAS in this study. Future research should aim to investigate this effect with a larger number of test participants.

## 5. Conclusions

In conclusion, our study demonstrated that the intake of *Helianthus tuberosus* in the morning under free-living conditions is a more effective way to suppress postprandial and 24 h glucose levels and improve the intestinal environment.

## Figures and Tables

**Figure 1 nutrients-12-03035-f001:**
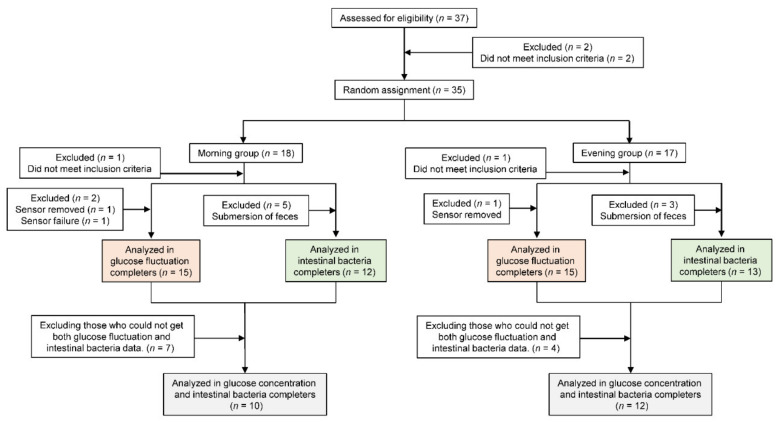
Consolidated Standards of Reporting Trials flow diagram.

**Figure 2 nutrients-12-03035-f002:**
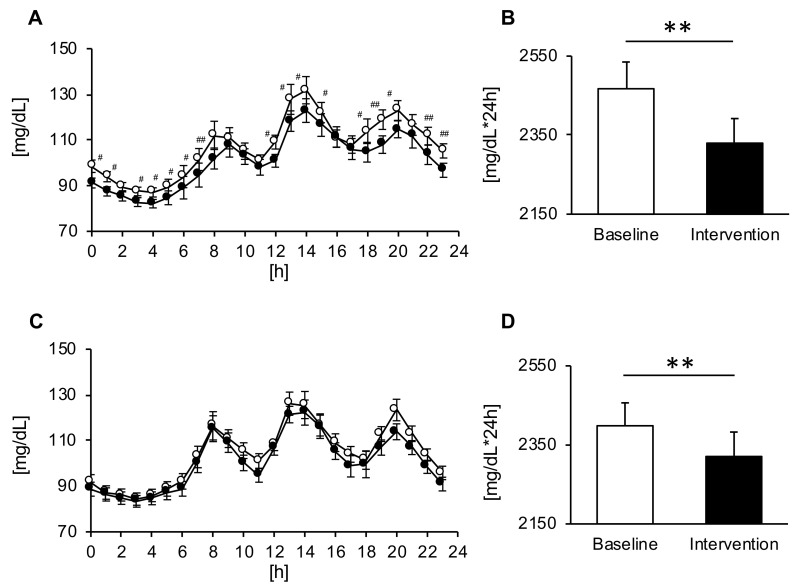
Diurnal changes in the blood glucose levels and area under the curve (AUC) across the day in the morning ((**A**,**B**), *n* = 15) and evening groups ((**C**,**D**), *n* = 15). Values are expressed as means and standard errors. ^#^
*p* < 0.05, ^##^
*p* < 0.01 compared to the baseline level (Bonferroni correction was used for post-hoc comparisons), ** *p* < 0.01 compared to the baseline level (paired *t*-test).

**Figure 3 nutrients-12-03035-f003:**
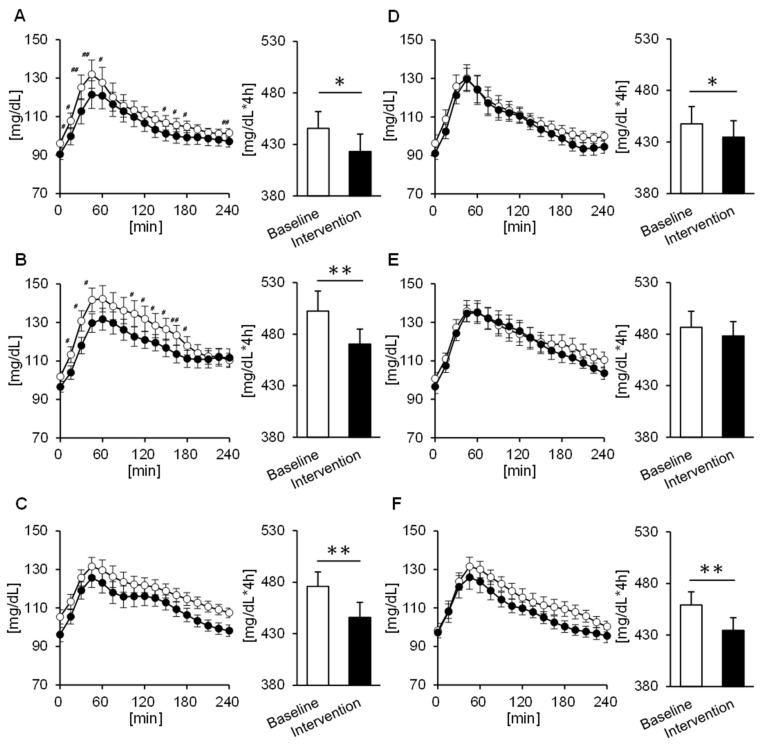
Concentrations of glucose 4 h postprandial and the area under the curve (AUC) in the morning group at (**A**) breakfast, (**B**) lunch, and (**C**) dinner, as well as in the evening group at (**D**) breakfast, (**E**) lunch, and (**F**) dinner. Participants in the morning group took Helianthus tuberosus just before breakfast, and participants in the evening group took it just before dinner. Values are expressed as means and standard errors. ^#^
*p* < 0.05, ^##^
*p* < 0.01 compared to the baseline level (Bonferroni correction for post-hoc comparisons), * *p* < 0.05, ** *p* < 0.01 compared to the baseline level (paired *t*-test).

**Figure 4 nutrients-12-03035-f004:**
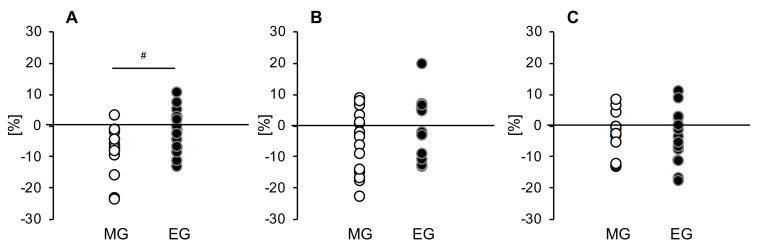
Comparison of the change rates of the peak blood glucose levels after each meal in both groups at (**A**) breakfast, (**B**) lunch, and (**C**) dinner. MG: morning group, EG: evening group. Values are expressed as means and standard errors. ^#^
*p* < 0.05 compared to the level in the evening group (unpaired *t*-test).

**Figure 5 nutrients-12-03035-f005:**
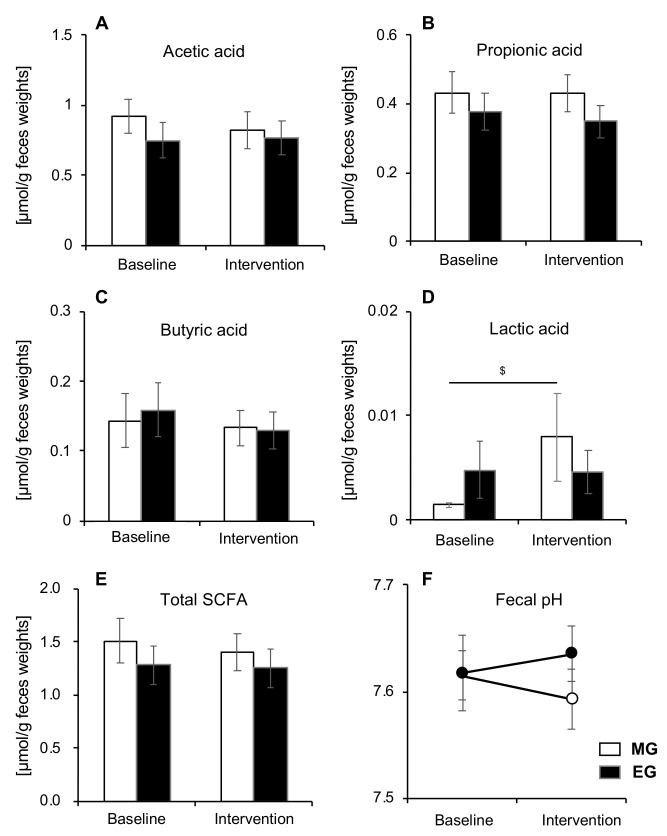
Changes in SCFA and pH before and after intervention in both groups (*n* = 22). (**A**) acetic acid, (**B**) propionic acid, (**C**) butyric acid, (**D**) lactic acid, (**E**) total SCFA, and (**F**) fecal pH. Values are expressed as means and standard errors. ^$^
*p* < 0.05 compared to baseline levels (Wilcoxon). SCFA: short-chain fatty acid, MG: morning group, EG: evening group.

**Figure 6 nutrients-12-03035-f006:**
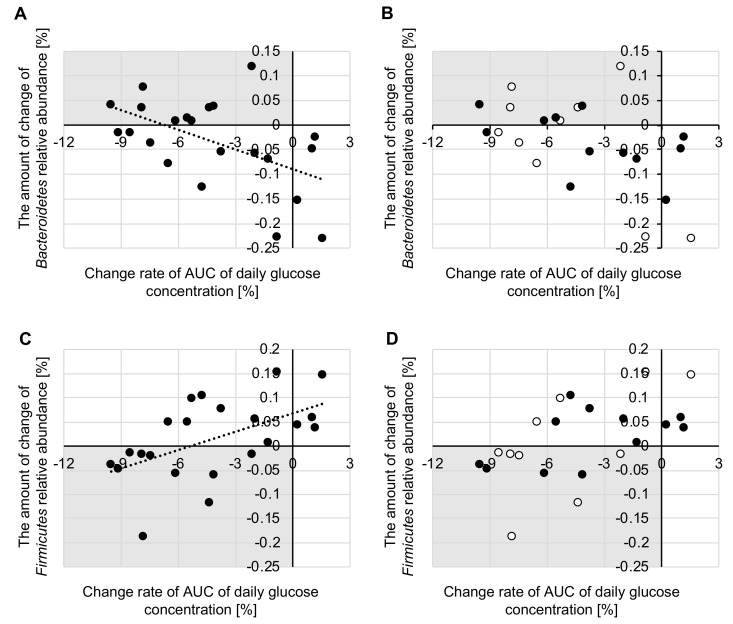
Correlation between the change rate of the area under the curve (AUC) for the diurnal glucose concentration and the amount of change of (**A**,**B**) phylum Bacteroidetes (**C**,**D**) relative abundance and Phylum Firmicutes (*n* = 22). The white plot indicates the morning group (**B**,**D**). The black plot indicates the evening group (**B**,**D**). (**A**) and (**C**) are the correlations for all participants. (**B**) and (**D**) are the correlations between each group.

**Figure 7 nutrients-12-03035-f007:**
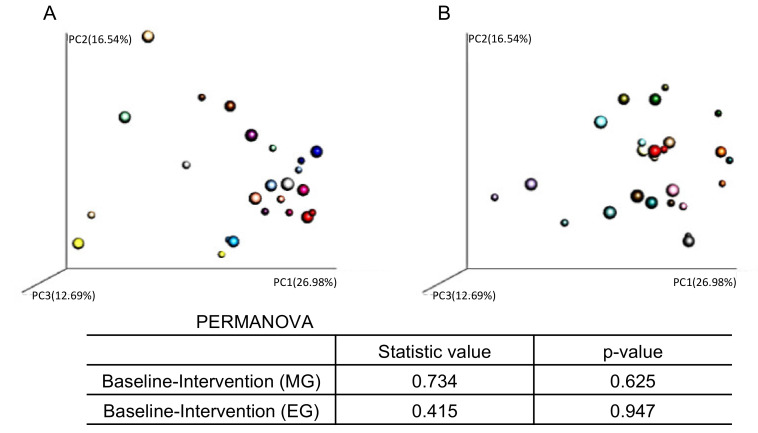
Individual comparison of fecal microbiome β-diversity (weighted UniFrac) in the (**A**) morning group and (**B**) evening group. The small plot indicates the baseline results. The large plot indicates the results after the intervention.

**Figure 8 nutrients-12-03035-f008:**
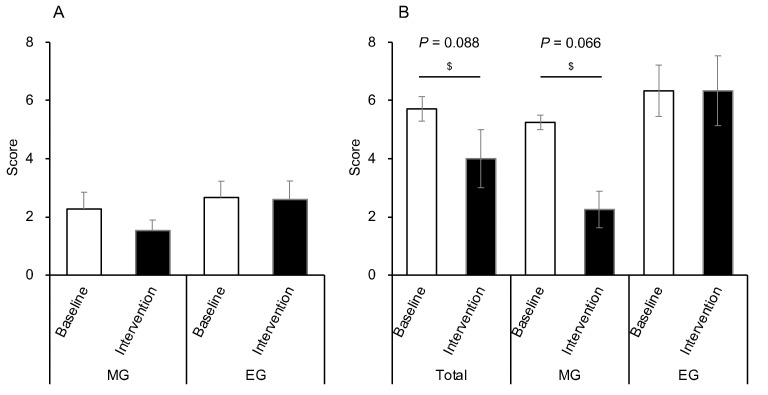
Changes in CAS before and after intervention in both groups. (**A**) Changes in CAS before and after intervention in all participants in both groups (MG: *n* = 15, EG: *n* = 15). (**B**) Changes in participants with constipation before and after the intervention (total: *n* = 7, MG: *n* = 4, EG: *n* = 3). Values are expressed as means and standard errors. ^$^
*p* < 0.05 compared to the baseline level (Wilcoxon). CAS: constipation assessment scale, MG: morning group, EG: evening group.

**Table 1 nutrients-12-03035-t001:** Baseline characteristics of study participants.

Physical Characteristics and Dietary Intake	All Participants (*n* = 30)	MG (*n* = 15)	EG (*n* = 15)
Age (years)	74.4 ± 1.0	74.3 ± 0.9	74.6 ± 1.7
Height (cm)	158.4 ± 1.6	158.1 ± 2.0	158.7 ± 2.6
Body Mass (kg)	55.3 ± 1.9	54.9 ± 2.8	55.7 ± 2.6
BMI (kg/m^2^)	21.8 ± 0.4	21.7 ± 0.7	21.9 ± 0.6
Muscle Mass (kg)	21.6 ± 0.9	21.3 ± 1.2	21.8 ± 1.3
MEQ	54.2 ± 1.3	53.0 ± 2.2	55.4 ± 1.5
Energy Intake (kcal/day)	2379.7 ± 134.7	2292.6 ± 112.9	2466.9 ± 247.7
Carbohydrate Intake (g/day)	289.2 ± 14.4	288.7 ± 14.5	289.7 ± 25.6
Fat Intake (g/day)	85.4 ± 7.2	81.6 ± 5.0	89.3 ± 13.6
Protein Intake (g/day)	87.8 ± 5.0	85.7 ± 4.3	89.9 ± 9.1
Total Dietary Fiber Quantity (g/day)	18.0 ± 1.3	17.9 ± 1.3	18.1 ± 2.3
Water Soluble Dietary Fiber Quantity (g/day)	4.0 ± 0.2	4.0 ± 0.3	4.1 ± 0.4
Insoluble Dietary Fiber Quantity (g/day)	12.6 ± 1.0	12.5 ± 0.9	12.7 ± 1.8

Note. All data are presented as the mean ± standard error. BMI: body mass index, MEQ: morningness-eveningness questionnaire, MG: morning group, EG: evening group.

**Table 2 nutrients-12-03035-t002:** Changes in glucose parameters in both groups.

Glucose Parameters	MG (*n* = 15)		EG (*n* = 15)	
	Baseline	Intervention	Baseline	Intervention
SD (mg/dL)	15.9 ± 1.6	15.3 ± 1.5	16.3 ± 1.3	15.2 ± 1.2
CV (%)	14.6 ± 1.2	15.0 ± 1.3	15.8 ± 1.3	15.3 ± 1.3
Max (mg/dL)	141.4 ± 5.7	134.5 ± 5.5	143.9 ± 5.1	135.0 ± 5.1
Min (mg/dL)	84.7 ± 1.8	80.1 ± 2.0	83.0 ± 2.7	81.7 ± 2.5
MAGE (mg/dL)	56.6 ± 5.2	54.4 ± 5.3	60.9 ± 5.0	53.3 ± 4.6

Note. All data are presented as the mean ± standard error. SD: standard deviation, CV: coefficient of variation, MAGE: mean amplitude of glycemic excursion, Max: maximum glucose, Min: minimum glucose, MG: morning group, EG: evening group.

**Table 3 nutrients-12-03035-t003:** Changes in the relative abundance of some bacteria in both groups (genus level).

Bacterial	MG (*n* = 10)		EG (*n* = 12)	
	Baseline	Intervention	Baseline	Intervention
*Bifidobacterium*	0.0592 ± 0.0189	0.0790 ± 0.0280	0.0902 ± 0.0255	0.0871 ± 0.0209
*Bacteroides*	0.1302 ± 0.0377	0.1287 ± 0.0385	0.1778 ± 0.0293	0.1267 ± 0.0198
*Parabacteroides*	0.0162 ± 0.0028	0.0169 ± 0.0045	0.0153 ± 0.0046	0.0159 ± 0.0064
*Lactobacillus*	0.0108 ± 0.0047	0.0152 ± 0.0076	0.0361 ± 0.0239	0.0296 ± 0.0160
*Streptococcus*	0.0299 ± 0.0116	0.0537 ± 0.0245	0.0497 ± 0.0202	0.0414 ± 0.0140
*Coprococcus*	0.0161 ± 0.0053	0.0137 ± 0.0062	0.0141 ± 0.0041	0.0158 ± 0.0043
*Dorea*	0.0098 ± 0.0022	0.0121 ± 0.0027	0.0129 ± 0.0029	0.0154 ± 0.0045
*Roseburia*	0.0086 ± 0.0029	0.0084 ± 0.0047	0.0087 ± 0.0033	0.0121 ± 0.0034
[*Ruminococcus*]	0.0068 ± 0.0022	0.0095 ± 0.0024	0.0127 ± 0.0044	0.0106 ± 0.0041
*Oscillospira*	0.0097 ± 0.0020	0.0081 ± 0.0020	0.0138 ± 0.0028	0.0087 ± 0.0018
*Ruminococcus*	0.0592 ± 0.0189 ^#^	0.0113 ± 0.0045 ^#^	0.0462 ± 0.0098	0.0523 ± 0.0115

All data are presented as the mean ± standard error. MG: morning group, EG: evening group. ^#^
*p* < 0.05 compared with the level in the EG (Bonferroni correction for post-hoc comparisons).

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
