# Peer review of "Ingestion of Helianthus tuberosus at Breakfast Rather Than at Dinner is More Effective for Suppressing Glucose Levels and Improving the Intestinal Microbiota in Older Adults"

_nutrients, 2020, doi:10.3390/nu12103035_

Round 1
Reviewer 1 Report
In this manuscript, the authors described that dietary intake of Helianthus tuberosus in the morning have relatively stronger effects on intestinal microbiota and suppress postprandial glucose levels than those in the evening.
Minor comments:
Line 303: A space is missing between the letter and Acetic Acid
Figures 7A, and B: are not quite clear. Please increase the quality of the PCoA graphs
Major comments:
Concerning the gut microbial analysis, did the authors investigate in a deeper manner the gut microbiota composition? They just mentioned the ratio between Firmicutes and Bacteroides but they have not shown any further taxonomic classification. It is quite important for this study to go deeper.
The different effects observed between the morning and the evening may be explained due to the circadian variation in two different timing of the day?
For the constipation assessment scale, it is not so clear the reason of having n=7 at the baseline, and n=4 in the group after intervention. Can the authors explain it?
The concentration of lactic acid in the feces is quite low, and there seems to be a big variation in the MG intervention group (MG), and no other changes have been observed in other SCFAs. I suggest authors to pay attention when describing the role of the SCFAs in the discussion. I really do not see a big change in the profile of those metabolites. They also mentioned the role exerted by GLP-1, but they did not measure it in this study. I highly ask the authors to clarify this latter point.
Author Response
We wish to thank the editor and reviewers for reading our manuscript so thoroughly and providing such constructive feedback. The quality of our manuscript has certainly improved as a result of these comments. Our responses are provided below in a point-by-point fashion and any changes in the revised manuscript are highlighted in yellow.
Reviewer 1:
In this manuscript, the authors described that dietary intake of Helianthus tuberosus in the morning have relatively stronger effects on intestinal microbiota and suppress postprandial glucose levels than those in the evening.
Query 1:
Line 303: A space is missing between the letter and Acetic Acid
Response 1:
Thank you for pointing out this mistake. We have revised the Figure 5 legend in the manuscript (lines 343–344).
Query 2:
Figures 7A, and B: are not quite clear. Please increase the quality of the PCoA graphs
Response 2:
As you suggested, we have increased the quality of the PCoA graphs (lines 377–378).
Query 3:
Concerning the gut microbial analysis, did the authors investigate in a deeper manner the gut microbiota composition? They just mentioned the ratio between Firmicutes and Bacteroides but they have not shown any further taxonomic classification. It is quite important for this study to go deeper.
Response 3:
As you suggested, we have added further genus levels in both groups (lines 367–374 and Table 3). We evaluated changes in the relative abundance of some bacteria in both groups (genus level). As a result, the relative abundance of Ruminococcus in the morning group at the genus level was significantly higher at baseline than in the evening group, and significantly lower after the intervention. However, no significant change was observed in the other bacteria. Also, the change rate of the AUC of the daily glucose level was not correlated with the relative abundance of bacteria.
Query 4:
The different effects observed between the morning and the evening may be explained due to the circadian variation in two different timing of the day?
Response 4:
Thank you for your comments. Glucose tolerance has been reported to be regulated by a circadian system, including time of day variations in digestion, absorption, and metabolism in the stomach and intestines 1,2). There is time-of-day variation in glucose tolerance, including insulin function, with maximum tolerance in the morning and minimum tolerance in the evening and night 3). Postprandial glucose levels have been suggested to be higher in the evening than in the morning. Furthermore, a previous study has shown that circadian variations are also present in the intestinal microbiota and are controlled by dietary composition. Therefore, we believe that there is an effect of circadian variation. Although several factors influence postprandial glucose metabolism, the effects of timing of anti-diabetes functional food intake on postprandial glucose levels and intestinal microbiota in humans are still unclear. We have added in this issue in the Introduction (lines 54–58, 77–78) and Discussion (lines 425–431).
- Pan, X.; Hussain, M.M. Clock is important for food and circadian regulation of macronutrient absorption in mice. J Lipid Res 2009, 50, 1800-1813, doi:10.1194/jlr.M900085-JLR200.
- Tavakkolizadeh, A.; Ramsanahie, A.; Levitsky, L.L.; Zinner, M.J.; Whang, E.E.; Ashley, S.W.; Rhoads, D.B. Differential role of vagus nerve in maintaining diurnal gene expression rhythms in the proximal small intestine. J Surg Res 2005, 129, 73-78, doi:10.1016/j.jss.2005.05.023.
- Lindgren, O.; Mari, A.; Deacon, C.F.; Carr, R.D.; Winzell, M.S.; Vikman, J.; Ahrén, B. Differential islet and incretin hormone responses in morning versus afternoon after standardized meal in healthy men. J Clin Endocrinol Metab 2009, 94, 2887-2892, doi:10.1210/jc.2009-0366.
Query 5:
For the constipation assessment scale, it is not so clear the reason of having n=7 at the baseline, and n=4 in the group after intervention. Can the authors explain it?
Response 5:
Thank you for your comments. Of all the participants in this study, seven were diagnosed with constipation (n=4 in the morning group and n=3 in the evening group) (lines 385–386).
As you pointed out, it may be difficult to explain the effect of Helianthus tuberosus powder intake on the Constipation Assessment Scale with this sample size. However, the authors believe that some improvement was seen, given similar results to the intestinal microbiota. Of course, it will be necessary to increase the sample size and examine it in more detail in the future. We have added in this issue in the Discussion (lines 535–536).
Query 6:
The concentration of lactic acid in the feces is quite low, and there seems to be a big variation in the MG intervention group (MG), and no other changes have been observed in other SCFAs. I suggest authors to pay attention when describing the role of the SCFAs in the discussion. I really do not see a big change in the profile of those metabolites. They also mentioned the role exerted by GLP-1, but they did not measure it in this study. I highly ask the authors to clarify this latter point.
Response 6:
Thank you for your comments. As you pointed out, we agree that attention should be paid to these interpretations in this study. Therefore, the relevant part was toned down and rewritten (lines 462–464, 466–472).
Reviewer 2 Report
The authors Kim et al. have investigated the effect of the timing of daily Helianthus tuberosus ingestion on postprandial and 24-h glucose levels and intestinal microbiota in older adults. To do so, they have recruited thirty-seven older adults, divided them to two groups based on H.T supplementation. They monitored glucose levels and performed 16S rRNA seq to understand the changes associated with the microbiome.
Comments:
- The major limitations of the study are recruiting the subjects from single site/hospital. It would have been beneficial to have subjects cross sites.
- Major strengths are use of older adults, although the authors have mentioned healthy older adults as a limitation, however, it can be considered as a strength as there is not much research conducted on the health aging.
- From the 16S seq and SCFAs results, it is clear that HT supplementation was not able to bringout much changes in the microbiome nor its associated SCFAs in either of the times, a small change in the morning group. Could that be inconsistency in supplementation ? Perhaps, 2 weeks of supplementation may not be enough to see changes. Please explain.
Question:
- The authors have used almost equal number of females:males in the study, did the authors find any differences between sexes? It is a great opportunity and worth analyzing.
- Did the authors accoun for any diary intake/yogurt/milk/cheese products from the recruited subjects, which may confound with the results.
- How did the authors choose 5grams of powder?
- Figure 5 can be labelled more appropriately with the names of SCFAs near the figure as well.
- The author's may well think about looking into differences in males/females in terms of glucose levels as well as microbiome, if there is any difference.
Author Response
We wish to thank the editor and reviewers for reading our manuscript so thoroughly and providing such constructive feedback. The quality of our manuscript has certainly improved as a result of these comments. Our responses are provided below in a point-by-point fashion and any changes in the revised manuscript are highlighted in yellow.
Reviewer 2:
The authors Kim et al. have investigated the effect of the timing of daily Helianthus tuberosus ingestion on postprandial and 24-h glucose levels and intestinal microbiota in older adults. To do so, they have recruited thirty-seven older adults, divided them to two groups based on H.T supplementation. They monitored glucose levels and performed 16S rRNA seq to understand the changes associated with the microbiome.
Query 1 (Comment 1):
The major limitations of the study are recruiting the subjects from single site/hospital. It would have been beneficial to have subjects cross sites.
Response 1:
Thank you for your comment, we absolutely agree. We also think that we should conduct an additional study in another population, such as in young adults and diabetes patients.
Query 2 (Comment 2):
Major strengths are use of older adults, although the authors have mentioned healthy older adults as a limitation, however, it can be considered as a strength as there is not much research conducted on the health aging.
Response 2:
Thank you for your comment. We have added this issue to the limitations (lines 525–527).
Query 3 (Comment 3):
From the 16S seq and SCFAs results, it is clear that HT supplementation was not able to bringout much changes in the microbiome nor its associated SCFAs in either of the times, a small change in the morning group. Could that be inconsistency in supplementation? Perhaps, 2 weeks of supplementation may not be enough to see changes. Please explain.
Response 3:
Thank you for your comment. In this study, 100% of the participants took the correct amount of Helianthus tuberosus powder. Therefore, it is unlikely that changes in the microbiome and related SCFAs were not seen because of uneven intake of Helianthus tuberosus powder.
Regarding the powder intake period, previous studies have revealed that inulin intake for 14 to 16 days was effective for examining blood glucose fluctuations and the microbiome [1,2]. Also, another study has shown that the intake of dietary fiber for 7 days changes the intestinal environment in humans [3]. Since we examined the effects of one week of intake of Helianthus tuberosus powder containing inulin, larger changes may be seen in longer-term studies (lines 116–118, 262–263).
[1] Tuohy KM, Finlay RK, Wynne AG, Gibson GR. A human volunteer study on the prebiotic effects of HP-inulin – faecal bacteria enumerated using fluorescent in situ hybridization (FISH). Anaerobe 2001; 7: 113-118.
[2] Ramirez-Farias C, Slezak K, Fuller Z, et al. Effect of inulin on the human gut microbiota: stimulation of Bifidobacterium adolescentis and Faecalibacterium prausnitzii. Br J Nutr 2008: 1-10.
[3] Bouhnik Y et al., The capacity of short-chain fructo-oligosaccharides to stimulate faecal bifidobacteria: a dose-response relationship study in healthy humans. Nutr J. 2006, 5:8. doi: 10.1186/1475-2891-5-8.
Query 4 (Question 1 & 5):
1.The authors have used almost equal number of females:males in the study, did the authors find any differences between sexes? It is a great opportunity and worth analyzing.
5.The author's may well think about looking into differences in males/females in terms of glucose levels as well as microbiome, if there is any difference.
Response 4:
Thank you for your suggestion. We compared males and females and added the relevant information (lines 277–281, 301–308, 328–330, 369–371, and 446–455).
For blood glucose fluctuations, the results for males were similar to the overall results, but no clear differences were seen for females. Regarding the microbiome, no significant differences were observed in neither males nor females due to the intake of Helianthus tuberosus powder.
Query 5 (Question 2):
Did the authors accoun for any diary intake/yogurt/milk/cheese products from the recruited subjects, which may confound with the results.
Response 5:
Thank you for your comment. The effects of dairy intake cannot be denied. However, according to the Food Frequency Questionnaire (FFQ), dairy intake such as yogurt, milk, and cheese products at baseline were similar in both groups. In addition, although there were no restrictions on the intake of dairy products, participants were asked to maintain the same daily life and were asked to consume similar diet contents as much as possible during the study. Therefore, we consider the effect of the ingestion of dairy products on the results to be small (lines 261–262).
Query 6 (Question 3):
How did the authors choose 5grams of powder?
Response 6:
Thank you for your comments. In previous studies, an intake of dietary fiber of approximately 5–20 g/day was used [1]. Therefore, in this study, the powder intake was set to 5 g in consideration of reducing the burden on the elderly participants (lines 120–123).
[1] Kelly G. Inulin-type prebiotics: a review. (Part 2). Altern Med Rev. 2009;14(1):36-55.
Query 7 (Question 4):
Figure 5 can be labelled more appropriately with the names of SCFAs near the figure as well.
Response 7:
As you suggested, we have relabeled Figure 5 with the names of the SCFAs (lines 331–342).

Reviewer 3 Report
The authors have examined the effect of timing of Helianthus tuberosus ingestion on postprandial and 24h glucose levels and intestinal microbiota in older adults. The study is informative, well structured, supported through pertinent studies from literature and novel in the sense that the previous studies have focussed on daily intake rather than considering the timing aspect. Keeping in view the novelty and technical soundness of the article, it will contribute to the advancement in scientific knowledge and will be of considerable interest for the readers. The detailed comments described for this article can be considered as “minor revisions” and given in different parts as per sequence of the paper to be considered for publication.
Abstract:
It would be nice to revise the abstract by adding principal results of key parameters e.g. results of baseline characteristics, changes in glucose parameters etc. It will be more informative and interesting for the readers.
Key words:
Seem to be repetition of most of the words appearing in the title of this article which should be replaced with appropriate words.
Introduction:
The information given in introduction is clear, concise and in sequence as far as technical soundness is concerned. They have started with prevalence of diabetes around the globe, its relationship with postprandial hyperglycemia and cardiovascular disease and highlighted the importance of day timings for regulating glucose level. They have also highlighted the role of dietary fiber, its association with growth of bacteria and glucose metabolism and regulation.
However, it is suggested to 1) add few studies related to timing effect for other foods to regulate glucose metabolism. (2) highlight the nutritional significance of Helianthus tuberosus, also mentioning its common name for nonspecialized readers just like other plants where they have mentioned their common names e.g. burdock and chicory in line 60. The authors are also suggested to mention the other technical name commonly used for Helianthus tuberosus i.e. Jerusalem artichoke.
Materials and methods:
The experimental design is appropriate and most of the methods are well conceived and described in an effective way. Further, all these methods are supported with the help of scientific references and international norms where required. However, the methods 2.4.8 and 2.5 seem too lengthy and can be further summarized.
Results and discussion:
The results have been presented and discussed in a scientific way. The authors have documented their results through graphical representation with the support of tables for discussing different aspects of pertinent results obtained during their study. The statistical methods and approach to discuss these results along with comparison with literature are appropriate. Only few grammatical and English language errors need to be corrected.
Conclusion:
Conclusion needs to be improved by adding pertinent results as suggested for abstract. Further, it is suggested to propose one or two strategies to effectively implement this diet and timing plan for elderly people on daily basis as prospects of the study.
Specific comments include:
Authors are suggested to go through the English language and grammar of the manuscript. Few examples are being mentioned here for their convenience.
Line 74: Add “conducted” after “This study was”.
Line 75-76: 1) not on any antioxidant….2) no diagnosis…. It will be more appropriate to use verbs and nouns for describing these sentences instead of just mentioning not or no at start of these criteria.
Line 96: “not change lifestyle…” can be replaced with “without changing lifestyle…..”.
Line 99: “in water or hot water”. Here, water stands for normal/cold water? If this is the case, it’s better to mention here.
Line 102: “and food frequency survey, was evaluated”, replace was with were.
Line 105: They were asked to collect their feces to the tube…Here the word “to” should be replaced with “in”.
Based on few examples, it is suggested that authors should improve the English language by consulting native English speaker.
Author Response
We wish to thank the editor and reviewers for reading our manuscript so thoroughly and providing such constructive feedback. The quality of our manuscript has certainly improved as a result of these comments. Our responses are provided below in a point-by-point fashion and any changes in the revised manuscript are highlighted in yellow.
Reviewer 3:
The authors have examined the effect of timing of Helianthus tuberosus ingestion on postprandial and 24h glucose levels and intestinal microbiota in older adults. The study is informative, well structured, supported through pertinent studies from literature and novel in the sense that the previous studies have focussed on daily intake rather than considering the timing aspect. Keeping in view the novelty and technical soundness of the article, it will contribute to the advancement in scientific knowledge and will be of considerable interest for the readers. The detailed comments described for this article can be considered as “minor revisions” and given in different parts as per sequence of the paper to be considered for publication.
Query 1:
Abstract:
It would be nice to revise the abstract by adding principal results of key parameters e.g. results of baseline characteristics, changes in glucose parameters etc. It will be more informative and interesting for the readers.
Response 1:
As you suggested, we have added these results in the Abstract (lines 27–28 and 29 –35).
Query 2:
Key words:
Seem to be repetition of most of the words appearing in the title of this article which should be replaced with appropriate words.
Response 2:
Thank you for your suggestion. We have replaced some of the words with different keywords (lines 42–43).
Query 3:
Introduction:
The information given in introduction is clear, concise and in sequence as far as technical soundness is concerned. They have started with prevalence of diabetes around the globe, its relationship with postprandial hyperglycemia and cardiovascular disease and highlighted the importance of day timings for regulating glucose level. They have also highlighted the role of dietary fiber, its association with growth of bacteria and glucose metabolism and regulation.
However, it is suggested to 1) add few studies related to timing effect for other foods to regulate glucose metabolism. (2) highlight the nutritional significance of Helianthus tuberosus, also mentioning its common name for nonspecialized readers just like other plants where they have mentioned their common names e.g. burdock and chicory in line 60. The authors are also suggested to mention the other technical name commonly used for Helianthus tuberosus i.e. Jerusalem artichoke.
Response 3:
Thank you for your suggestion. We have added some studies related to the timing effect of other food for the regulation of glucose metabolism in the Introduction (lines 79–81).
1.Takahashi, M.; Ozaki, M.; Miyashita, M.; Fukazawa, M.; Nakaoka, T.; Wakisaka, T.; Matsui, Y.; Hibi, M.; Osaki, N.; Shibata, S. Effects of timing of acute catechin-rich green tea ingestion on postprandial glucose metabolism in healthy men. J Nutr Biochem 2019, 73, 108221, doi:10.1016/j.jnutbio.2019.108221.
2.Takahashi, M.; Ozaki, M.; Tsubosaka, M., Kim, H.-K.; Sasaki, H.; Matsui, Y.; Hibi, M.; Osaki, N.; Miyashita, M.; Shibata, S. Effects of Timing of Acute and Consecutive Catechin Ingestion on Postprandial Glucose Metabolism in Mice and Humans. Nutrients 2020, 12, 565. doi: 10.3390/nu12020565.
Query 4:
Materials and methods:
The experimental design is appropriate and most of the methods are well conceived and described in an effective way. Further, all these methods are supported with the help of scientific references and international norms where required. However, the methods 2.4.8 and 2.5 seem too lengthy and can be further summarized.
Response 4:
As you suggested, we have further summarized in this part in the Methods (lines 205–224 and 241–257).
Query 5:
Results and discussion:
The results have been presented and discussed in a scientific way. The authors have documented their results through graphical representation with the support of tables for discussing different aspects of pertinent results obtained during their study. The statistical methods and approach to discuss these results along with comparison with literature are appropriate. Only few grammatical and English language errors need to be corrected.
Response 5:
Thank you for your comments. We have checked these points ourselves and have asked Editage (www.Editage.jp) to review the manuscript and provide English language editing.
Query 6:
Specific comments include:
Authors are suggested to go through the English language and grammar of the manuscript. Few examples are being mentioned here for their convenience.
Line 74: Add “conducted” after “This study was”.
Line 75-76: 1) not on any antioxidant….2) no diagnosis…. It will be more appropriate to use verbs and nouns for describing these sentences instead of just mentioning not or no at start of these criteria.
Line 96: “not change lifestyle…” can be replaced with “without changing lifestyle…..”.
Line 99: “in water or hot water”. Here, water stands for normal/cold water? If this is the case, it’s better to mention here.
Line 102: “and food frequency survey, was evaluated”, replace was with were.
Line 105: They were asked to collect their feces to the tube…Here the word “to” should be replaced with “in”.
Based on few examples, it is suggested that authors should improve the English language by consulting native English speaker.
Response 6:
Thank you for your comments. Thank you for pointing out these mistakes. We have asked Editage (www.Editage.jp) to carefully proofread and edit the manuscript to ensure that all grammatical errors are corrected (lines 92, 93–94, 114, 120, and 128).

Round 2
Reviewer 1 Report
Dear authors,
I really appreciate all the changes that have been done in accordance with my comments and suggestion. However, I still found that the manuscript needs to be improved and better detailed in some parts. Please see my comments below:
Line68: I would not use the term “good bacteria. Soma bacteria can be considered as good or bad depending on the pathological situations. So be careful with that kind of definition.
I will introduce more the definition of SCFAs (e.g., the end products of anaerobic bacterial fermentation..)
Lines 93-94: Did the authors consider either probiotics or antibiotics as exclusion criteria?
Line 362: For the correlation analysis: Did the authors calculate the adjusted p-value? If so, they need to report it.
Lines 408-409: As shown by the PCoA profile, Helianthus tuberosus supplementation was not able to really affect the gut microbiota composition. I kindly ask the authors to rewrite this part and to pay attention when drawing certain conclusions.
Lines 409-416: Same comment as above. Please pay attention when drawing a certain conclusion. You do not have a big difference in constipation.
Lines 463-473: The fact that you do not see the difference in the content of the SCFAs does not preclude that you have no differences in systemic circulation, and also certain SCFAs (i.e, Butyrate) can be used as energy substrate from the colonocytes. I would suggest the authors to include this part in the text.
Line 496: I would not use the word good and bad bacteria (see my comment above).
Lines 501-505: Authors wrote that: Our findings indicate that Helianthus tuberosus intake in the morning might have relatively stronger effects on the intestinal microbiota and suppress postprandial glucose levels to a greater extent than when taken in the evening.
At the same time, they said that our results showed that only the relative amount of Ruminococcus was higher before the intervention than in the evening intake group, but this amount was significantly lower after the intervention.
So, is it better to take Helianthus tuberosus supplementation in the morning or in the evening? Authors have to better explain this part.
There are other studies describing the role of gut dysbiosis, gut barrier dysfunction, and low-grade inflammation. Reference 53-54 are not the right ones. Please check the literature and insert the original references.
Author Response
Paper No. Nutrients-926377
Response to the Reviewers’ comments:
We wish to thank the editor and reviewers for reading our manuscript so thoroughly and providing such constructive feedback. The quality of our manuscript has certainly improved as a result of these comments. Our responses are provided below in a point-by-point fashion and any changes in the revised manuscript are highlighted in yellow.
Reviewer 1:
I really appreciate all the changes that have been done in accordance with my comments and suggestion. However, I still found that the manuscript needs to be improved and better detailed in some parts. Please see my comments below:
Query 1:
Line68: I would not use the term “good bacteria. Soma bacteria can be considered as good or bad depending on the pathological situations. So be careful with that kind of definition.
Response 1:
Thank you for your comments. We have rewritten this sentence (lines 67–69).
Query 2:
I will introduce more the definition of SCFAs (e.g., the end products of anaerobic bacterial fermentation..)
Response 2:
As you suggested, we have added more information to the definition of SCFAs (lines 67–69).
Query 3:
Lines 93-94: Did the authors consider either probiotics or antibiotics as exclusion criteria?
Response 3:
Thank you for your comment. The effects of probiotics or antibiotics cannot be denied. However, according to the Food Frequency Questionnaire (FFQ), total dietary fiber (includes water-soluble dietary fiber and insoluble dietary fiber) at baseline was similar in both groups. In addition, all participants were asked to maintain the same daily life and were asked to consume similar diet contents as much as possible during the study. Also, at least during the study, no participants become ill, so it is presumed that they did not take antibiotics. Therefore, we consider the effect of the ingestion of probiotics or antibiotics on the results to be small (lines 115–117).
Query 4:
Line 362: For the correlation analysis: Did the authors calculate the adjusted p-value? If so, they need to report it.
Response 4:
Thank you for your comments. We did not use adjusted P-value and do not need to be reported.
Query 5:
Lines 408-409: As shown by the PCoA profile, Helianthus tuberosus supplementation was not able to really affect the gut microbiota composition. I kindly ask the authors to rewrite this part and to pay attention when drawing certain conclusions.
Response 5:
Thank you for your comments. We have rewritten this sentence (lines 408–410).
Query 6:
Line 496: I would not use the word good and bad bacteria (see my comment above).
Response 6:
Thank you for your comments. We have rewritten this sentence (lines 494–495).
Query 7:
Lines 501-505: Authors wrote that: Our findings indicate that Helianthus tuberosus intake in the morning might have relatively stronger effects on the intestinal microbiota and suppress postprandial glucose levels to a greater extent than when taken in the evening.
At the same time, they said that our results showed that only the relative amount of Ruminococcus was higher before the intervention than in the evening intake group, but this amount was significantly lower after the intervention.
So, is it better to take Helianthus tuberosus supplementation in the morning or in the evening? Authors have to better explain this part.
Response 7:
Thank you for your comments. We have rewritten this issue in the discussion (lines 497–502), and we deleted discussion [previous lines 500-507] about Ruminococcus including ref #53-54.
Query 8:
There are other studies describing the role of gut dysbiosis, gut barrier dysfunction, and low-grade inflammation. Reference 53-54 are not the right ones. Please check the literature and insert the original references.
Response 8:
Thank you for your comments. We have removed the sentences including references (53-54).
